# Effects of horizontal displacement and inter-character spacing on transposed-character effects in same-different matching

**Stéphanie Massol**[1]*, **Jonathan Grainger**[2,3]

**1** Laboratoire d'Étude des Mécanismes Cognitifs (EA 3082), Université Lumière Lyon 2, Lyon, France,
**2** Laboratoire de Psychologie Cognitive, Aix-Marseille University & CNRS, Marseille, France, **3** Institute of Language, Communication and the Brain, Aix-Marseille University, Marseille, France

* stephanie.massol@univ-lyon2.fr

**Data Availability Statement:** Data availability The authors confrm the availability of shared data. The datasets are accessible on the OSF website at: https://osf.io/s4jv7/.

## Abstract

In two same-different matching experiments we investigated whether transposed-character effects can be modulated by the horizontal displacement or inter-character spacing of target stimuli (strings of 6 consonants, digits, or symbols). Reference and target stimuli could be identical or differed either by transposing or substituting two characters. Transposition costs (greater difficulty in detecting a difference with transpositions compared with substitutions) were greater for letter stimuli compared to both digit and symbol stimuli in both experiments. In Experiment 1, half of the targets were displayed at the center of the screen and the other half were shifted by two character-positions to the left or to the right, whereas the reference was always presented at the center of the screen. Target displacement made the task harder and caused an increase in transposition costs whatever the type of stimulus. In Experiment 2, all stimuli were presented at the center of the screen and the inter-character spacing of target stimuli was increased by one character space on half of the trials. Increased spacing made the task harder and paradoxically caused an increase in transposition costs, but only significantly so for letter stimuli, and only in the discriminability (d') measure. These results suggest that target location and inter-character spacing manipulations caused an increase in positional uncertainty during the processing of location-specific complex features prior to activation of a location-invariant representation of character-in-string order. The hypothesized existence of a letter-specific order encoding mechanism accounts for the greater transposition costs seen with letter stimuli, as well as the greater modulation of these effects by an increase in inter-character spacing seen in discriminability (d').

## Introduction

One topic that has attracted a great deal of attention in the last decades is how the literate brain encodes the identities and positions of the constituent letters of words in written languages that use an alphabetic script. One of the major outcomes of these investigations is that the word recognition system is tolerant to small variations in letter position [1–7 see also 8].

**Funding:** JG was supported by ERC Grant 742141. The funders had no role in study design, data collection and analysis, decision to publish, or preparation of the manuscript.

**Competing interests:** The authors have declared that no competing interests exist.

Specifically, it has been demonstrated that it is still possible to read and to access the meaning of words in which letter order has been slightly modified (the so-called transposed-letter effect, e.g., JUGDE–JUDGE) [4–6, 9, 10]. Although previous studies contributed a great deal to the understanding of processes involved during the earliest stages of visual word recognition, debate continues with the respect to whether or not the mechanism used to code for letter position information in printed words is essentially the same mechanism as might be used to code for positional information in any kind of character string (e.g., digit strings and symbol strings).

Particularly relevant with respect to this debate, are studies providing evidence for greater transposition effects for letter strings compared with digit strings and symbol strings [11–13]. These studies used the same-different matching paradigm in which a reference stimulus is presented for 300 ms, immediately followed by a target stimulus for 300 ms, and participants are asked to judge whether or not the two stimuli are the same or not [14, 15]. This task can therefore be usefully applied to compare processing of different types of stimuli, such as letter, digit and symbol strings. Duñabeitia et al. (2012) [12] compared same-different judgments to strings of four characters that differed either by transposing two adjacent characters or by substituting the same two characters with different ones. These authors reported greater transposition costs (i.e., the difference between the transposition condition and the substitution condition) for letter strings compared with both digit and symbol strings. That is, participants needed more time and made more errors to decide that two strings differed by a character transposition (e.g., NDTF-NTDF) than to decide that two strings differed by a character substitution (e.g., NDTF-NSBF), and these transposed-character effects were significantly larger for letter strings than for the other types of character string. Massol et al. (2013) [13] replicated and extended these findings to 6-character strings.

The pattern of results obtained by Duñabeitia et al. (2012) and by Massol et al. (2013) [12, 13] is problematical for models that apply generic positional noise [16–18]. Indeed, these models assume that transposed-letter effects are a consequence of object position uncertainty as predicted by general models of visual attention [19, 20]. Specifically, transposed-letter effects are driven by generic positional noise that operate on an otherwise rigid position-coding mechanism. Crucially, this class of model [16–18] postulates that the processing of positional information in strings of unrelated letters (e.g., the consonant strings tested in the present study) is basically the same as the processing of positional information with strings of comparable non-letter stimuli such as digits and symbols (see [13], for a discussion). In support of this approach, when using the masked priming version of the same-different matching task, García-Orza, Perea, and Muñoz (2010) [21] found a highly similar transposed-character effect for letter, digit and symbol strings (see also [22]). Therefore, according to this approach, this apparent flexibility in the positional information encoding of letters, as reflected in transposed-letter effects, would be a by-product of a general property of the visual recognition system (i.e., positional noise). Moreover, other studies have shown that perceptual manipulations, such as highlighting the critical transposed-letters [23] reduce transposition effects. Nevertheless, as noted above, several studies using the same-different matching task without priming have revealed significantly larger transposition costs for letter strings than for both digit and symbol strings [12, 13]–a result that pleads in favor of a flexible position-coding scheme that is specific for letter strings [24–28]. In sum, given the evidence in favor of a role for relatively low-level perceptual factors plus the evidence for an impact of higher-level orthographic factors, we would argue that any complete account of transposed-character effects must involve both of these influences. This is the approach tested in the present work.

According to this theoretical approach, illustrated in Fig 1, transposed-letter effects are at least partly, but not uniquely, driven by a flexible letter-specific position coding mechanism

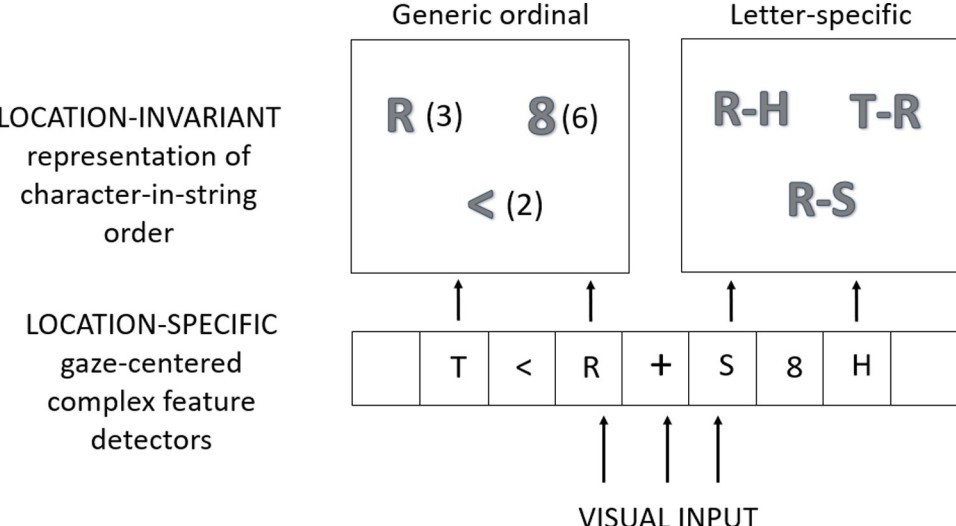

**Fig 1. Theoretical framework for the processing of identity and order information for same-different matching with strings of letters, digits, or symbols.** Stimuli are first encoded via location-specific gaze-centered complex feature detectors. Location-specific complex features then activate location-invariant object identities that are assigned an order in the string. Crucially, we hypothesize the existence of two different location-invariant order-encoding mechanisms: a generic order encoding mechanism, illustrated here by a simple ordinal representation (e.g., R(3), there is the letter R at the 3rd position), and a letter-specific order encoding mechanism (open-bigram coding in the examples shown in the upper-right panel of the Figure (e.g., T-R, there is a T before an R).

[27–30], inherent to the reading system. The backbone of this alternative approach was first proposed by Grainger and van Heuven (2004) [30] who introduced the key distinction between a location-specific (gaze-centered) encoding of the position of letter identities in a string of letters and a location-invariant (i.e., independent of viewing position) encoding of letter order (see Fig 1). At the first level of gaze-centered processing in the model shown in Fig 1, location-specific representations of complex features provide information that a given character is present in the stimulus array at a particular location relative to eye-fixation. These complex features include whole-letter and whole-digit representations as well as the complex features thought to be involved in identifying visual objects other than words and numbers. Processing at this level is subject to a certain amount of positional noise that increases with eccentricity [31]. Information about the location of complex features relative to eye-fixation then generates activity in a location-invariant representation of the identity of the different characters and their ordering. This two-level approach to position-coding is one key characteristic of our model that distinguishes it from the single-level approach adopted by the overlap model [16], for example.

The other key difference in our approach is that we hypothesize two different mechanisms for encoding location-invariant order information: a generic order encoding mechanism that operates independently of the Type of Stimulus (upper-left panel of Fig 1), and a letter-specific order encoding mechanism (upper-right panel of Fig 1). We illustrate this hypothesis by using a simple ordinal representation for the generic order encoding mechanism, and open-bigram coding [28, 30] for the letter-specific mechanism. Character transposition effects can be caused by positional noise operating on generic order encoding, such as in the overlap model [16]. Crucially, for letter stimuli, transposition effects can also be driven by the flexible nature of the letter-specific order encoding mechanism [see 26, 32, 33]. The additional role played by letter-specific order encoding in driving transposition effects obtained with letter strings explains

why these effects are greater than those obtained with both digit and symbol strings in the same-different matching task [12, 13].

It is important to note that the letter-specific location-invariant order encoding mechanism hypothesized in our model (i.e., open bigrams: an unordered set of ordered pairs of contiguous and non-contiguous letters) can account for transposed-letter effects without appealing to positional noise [30]. This means of encoding letter order information is hypothesized to provide a more efficient access to lexical representations on the basis of letter-level processing during reading [26], and is thought to develop during the course of learning to read [34–36]. This is hypothesized to be a reading-specific mechanism because of the very nature of the reading process whereby a given word identity can often be determined on the basis of knowledge of a subset of the word's letters and their relative positions. This is not the case for number processing, for example, where the very precise position of each digit is essential for obtaining accurate magnitude information (see [25], for further discussion).

In sum, according to the model described in Fig 1, positional noise would operate at the level of location-specific processing and would therefore apply to the processing of strings of different kinds of stimuli. On top of that, letter stimuli would additionally be impacted by the flexible nature of letter-specific order encoding, thus accounting for the greater transposition costs for letters compared with other kinds of familiar visual stimuli. Evidence for a perceptual locus for transposed-letter effects [21, 23, 37] does not counter this approach, given that our model includes both a perceptual (location-specific) and an orthographic (location-invariant) locus of such effects (see Fig 1). The key characteristic of our approach is that transposed-character effects in same-different matching are driven by both perceptual (i.e., location-specific processing–the 1st level in Fig 1), and higher-level location-invariant processing of order information (the 2nd level in Fig 1). The different pattern of findings across studies, discussed above, might therefore be due to the differences in the relative weighting assigned to location-specific and location-invariant representations when making same-different judgments as a function of the procedure that was employed.

In the present study we aimed to isolate effects of low-level perceptual noise and higher-level order-encoding processes on character-transposition effects in same-different matching. The use of the same-different matching task allowed us to investigate transposed-character effects with different types of stimuli (letters, digits, symbols). Crucially, we added physical manipulations of stimuli in order to investigate the impact of perceptual factors. More precisely, we investigated the influence of horizontal displacements of target stimuli on transposed-character effects in the same-different matching task (Experiment 1) and the influence of varying the inter-character spacing of target stimuli (Experiment 2). We hypothesized that these two manipulations would mainly affect processing of location-specific complex features (i.e., the first level of processing in Fig 1). This should therefore impact on all types of character strings and provides the first component contributing to a modulation of transposed-character effects. In addition to this, positional uncertainty at the level of location-invariant order encoding mechanisms provides a second component impacting on transposed-character effects. It is at the second level of order encoding that we hypothesize greater effects for letters compared with both digits and symbols, and a different impact of the inter-character spacing manipulation for the different types of stimuli. That is, although an increase in inter-character spacing should provide more accurate order information hence leading to smaller transposition effects driven by the generic ordinal code (top left panel of Fig 1), the more robust encoding of the relative positions of letters (top right panel of Fig 1) could lead to greater transposition effects given that it is the code itself (i.e., open-bigrams) that drives the effects.

In the present study participants were presented with reference-target pairs of 6-character strings that could be random strings of consonants (e.g., DKLNFT), digits (e.g., 349256) or

symbols (e.g., &+!?#$). The critical reference-target pairs could differ by transposing or substituting two non-contiguous characters that were always 1-character apart (e.g., KT̲D̲L̲NB–KLDTNB vs. KT̲D̲L̲NB–KF̲D̲G̲NB). In Experiment 1, the reference was always presented horizontally and aligned with the center of the screen, whereas the target stimulus could appear at the center of the screen or shifted by two character-positions to the left or to the right, and always one line below the reference stimulus. Statistical analyses were performed primarily on error rates and on discriminability indices (i.e., d' values) using Signal Detection Theory [38]. This choice was driven by the fact that quite high error rates are typical in same-different matching experiments with a brief presentation duration of the reference stimulus [see 12, 13], as in the present study. Therefore, error rates could be considered to be the main dependent variable. Furthermore, analyzing discriminability indices (d') allowed us to correct for potential response biases, hence revealing effects driven by perceptual processing as opposed to decision-level processes. However, given that the task instructions required participants to respond as rapidly as possible, we also analyzed RTs (see Supplemental materials).

## Experiment 1

### Methods

**Participants.** 71 participants (46 female) with a mean age of 24 (SD = 2.81) years took part in the experiment. They were paid for their collaboration. All of them were native speakers of Spanish and had normal or corrected-to-normal vision. All the participants signed informed consent forms before the experiment and were appropriately informed regarding the basic procedure of the experiment, according to the ethical commitments established by the BCBL Scientific Committee and by the BCBL Ethics Committee that approved the experiment.

**Materials.** 960 reference-target pairs were used as stimuli. Each of the pairs was composed of two 6-character long strings of uppercase consonants, digits, or symbols. These three categories consisted of 320 letter strings, 320 digit strings, or 320 symbol strings. For the digit strings, the numbers 1, 2, 3, 4, 5, 6, 7, 8 and 9 were used. For the letter strings, the uppercase version of the consonants, G, N, D, K, F, T, S, B and L were used. For the symbol strings, the characters %,?, !, &, +, <,), $ and # were used. In each category, half of the items required a "same" response (160 trials, i.e., 349256–349256, DKLNFT- DKLNFT, &+!?$#-&+!?$#). The other half (160 trials) required a "different" response. Half of the different pairs differed by means of character transpositions (transposed condition) or of character substitutions (substituted condition). The two transposed or substituted characters were always separated by one intervening character (i.e., 80 trials of transpositions, DK̲L̲N̲FT-DN̲L̲K̲FT; 80 trials including substitution with one intervening character, KT̲D̲L̲NB-KF̲D̲G̲NB, per category of character string). Transpositions or substitutions never involved outer characters and were equally distributed across the inner character positions. Furthermore, while reference stimuli were always presented at the center of the screen, half of the target stimuli were displayed in the center of the screen (i.e., 40 trials per category) and the other half were shifted by two characters to the left (i.e., 20 trials per category) or two characters to the right (i.e., 20 trials per category). In each list, each reference was presented twice, once requiring a "same" response and once requiring a "different" response, whereas each target occurred only once. A 3x2x2 within-participants factorial design was constructed, with Type of Stimulus (letter, digit, symbol), Target Location (i.e., central vs. shifted) and Type of Change (i.e., transposition vs. substitution) as main factors. Following a Latin-square design, the reference-target pairs were separated into eight experimental lists that were presented to different participants.

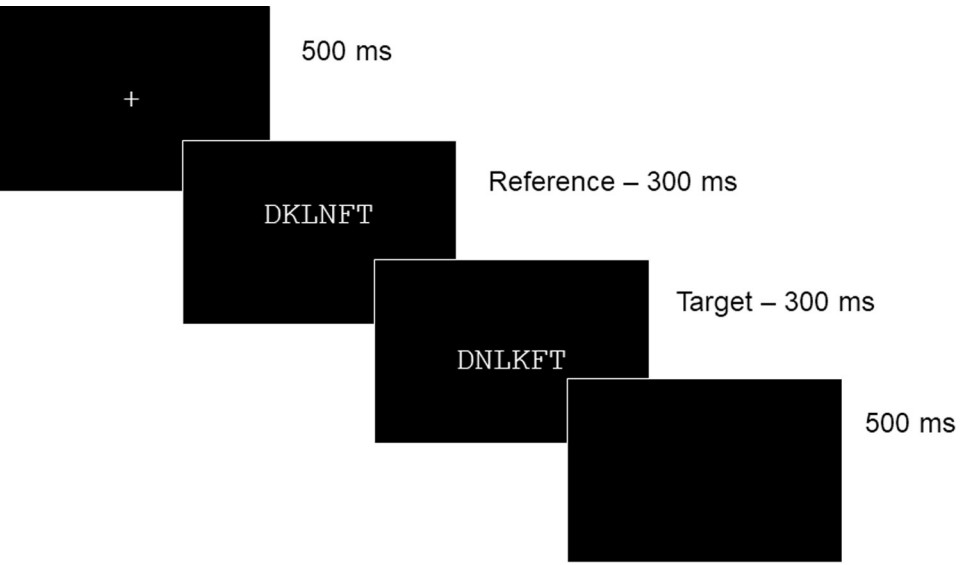

**Fig 2. Schematic representation of an experimental trial with an example of a "different" response for transposed-letter stimuli.**

**Procedure.** The presentation of the stimuli and recording of the responses were carried out using Presentation software. All stimuli were presented on a CRT monitor. Participants were informed that two strings of characters were going to be subsequently displayed. All stimuli were presented in white Courier New font (size 16 pt.) on a black background. Each trial began with the centered presentation of a fixation stimulus (*) displayed for 500ms. Immediately after this, the reference was presented for 300ms horizontally centered and positioned 3mm above the exact center of the screen. The reference was immediately replaced by the target stimulus that was positioned 3mm below the center of the screen. Target stimulus remained on the screen for 300ms. Each trial ended with participant's response, following by a blank screen displayed for 500ms. The manipulation of the location of references and targets on the vertical axis was carried out in order to avoid physical overlap between the two strings (see Fig 2 for a schematic representation of a trial). Participants were instructed to decide as rapidly and as accurately as possible whether or not the two strings were exactly identical. They responded "same" by pressing the "L" button on the keyboard and "different" by pressing the "S" button. Stimulus presentation was randomized and split into six blocks, with a short break between blocks. A short practice session was administered before the main experiment to familiarize participants with the procedure and the task.

## Results

Statistical analyses were performed only over the "different" trials, since there was no experimental manipulation of Type of change within the set of "same" trials. The dataset from one participant was excluded, based on an overall error rate above 40%. Our main analyses involved error rates and discriminability indices (d') obtained using Signal Detection Theory [38]. Analyses of RTs for both "same" and "different" responses are available in the Supplemental Materials (see S1 File).

Error rates were analyzed using generalized linear mixed-effects (GLME) models with items and participants as crossed random effects (including by-item and by-participant random intercepts [39] and with random slopes [40]. The model included Type of Character

(letter, digit, symbol), Target Location (central vs. shifted) and Type of Change (transposition vs. substitution) as fixed-factors. The model was fitted with the glmer function from the lme4 package [41] in the R statistical computing environment (version 4.1.0 [42]).

We also opted to analyze participants' discriminability indices (d') using Signal Detection Theory [38]. Compared with simple error rate analyses, discriminability corrects for response biases by combining hits (correct "different" responses) with false alarms (incorrect "different" responses). Discriminability (d') was calculated for each participant in the different experimental conditions. ANOVAs on the d' indices were conducted based on a 3 (Type of Stimulus: letter, digit, symbol) x 2 (Target Location: central vs. shifted by two character-positions to the left/right) x 2 (Type of Change: transposition vs. substitution) factorial design. The Greenhouse and Geisser (1959) [43] correction was applied to determine the significance values for variables with more than two levels.

**Error rates.** The maximal random effects structure that converged was one including by-participant random intercepts. The following analyses were conducted taking the letter string condition as reference for the Type of Character factor, the central condition as reference for the Target Location factor, and the substitution condition as reference for the Type of Change factor. We report regression coefficients (*b*), standard errors (*SE*) and z-values. Fixed effects were deemed significant if $|z| > 1.96$ [39]. Mean error rates for each of the experimental conditions are presented in Fig 3.

Error rates were lower for digit strings than for letter strings (28.95% vs. 40.78% respectively; $b = -0.49$, $SE = 0.06$, $z = -7.46$), whereas there was no significant difference between letter strings and symbol strings (40.78% vs. 39.66% respectively; $z = 1.21$). Fewer errors were observed when the target was presented at the center of the screen than when it was shifted by two characters to the left/right (30.46% vs. 42.47% respectively; $b = 0.48$, $SE = 0.06$, $z = 8.01$). There was also a significant effect of Type of Change, with fewer errors in the substitution condition than in the transposition condition (28.33% vs. 44.60% respectively; $b = 0.77$, $SE = 0.05$, $z = 12.98$). The interaction between Target Location and Type of Change was significant ($b = 0.30$, $SE = 0.08$, $z = 3.66$), reflecting the fact that the effect of Type of Change was greater in the displaced target condition (see Fig 3). However, the effect of Type of Change factor was significant in both the central target condition ($\chi^2(1) = 360.14$, $p < .001$) and the displaced target condition ($\chi^2(1) = 733.28$, $p < .001$). Finally, the interaction between Type of Character and Type of Change was significant ($b = -0.17$, $SE = 0.08$, $z = -2.07$). As can be seen in Fig 3, transposition effects were greater for letters compared with both digits and symbols, although the effects were significant for all types of character (letters: $\chi^2(1) = 497.65$, $p < .001$, digits: $\chi^2(1) = 302.29$, $p < .001$, symbols: $\chi^2(1) = 274.89$, $p < .001$). The other two-way interactions and the three-way interaction were not significant.

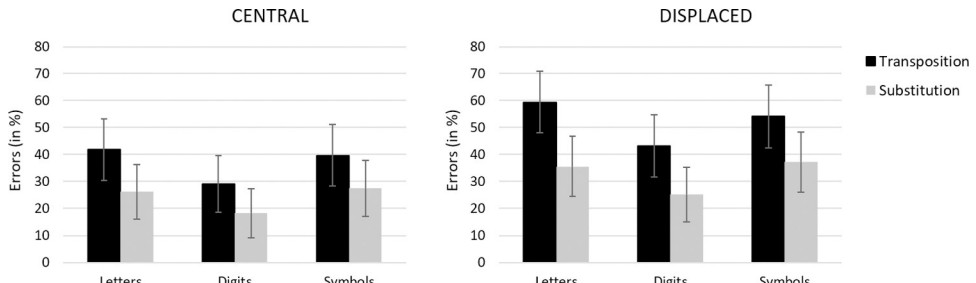

**Fig 3. Error rates for each type of stimulus in the transposition and in the substitution conditions when the target string was presented centrally (left panel) and when it was presented displaced two character spaces to the right or left (right panel) in Experiment 1.** Error bars represent within-participant 95% CIs [44].

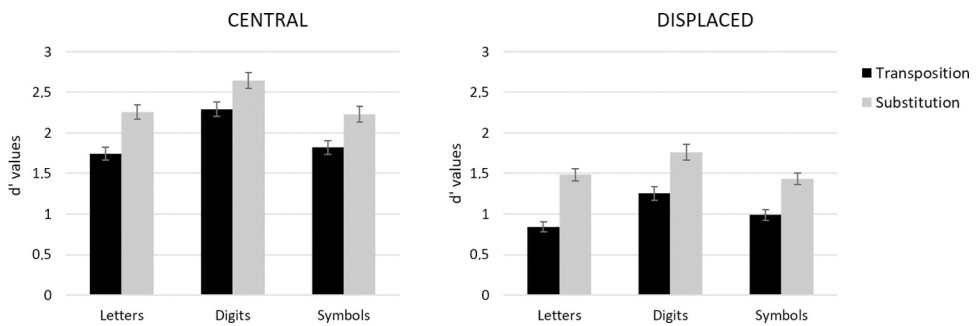

**Fig 4. d prime values for each type of stimulus in the transposition and in the substitution conditions when the target string was presented centrally (left panel) and when it was presented displaced two character spaces to the right or left (right panel) in Experiment 1.** Error bars represent standard errors.

**Discriminability.** Condition means are presented in Fig 4. There were main effects of Type of Stimulus, $F(2, 140) = 44.08$, $MSE = 14.32$, $p < .001$, $\eta_p^2 = .386$, Target Location, $F(1, 70) = 345.33$, $MSE = 161.83$, $p < .001$, $\eta_p^2 = .831$, and Type of Change, $F(1, 70) = 197.72$, $MSE = 48.69$, $p < .001$, $\eta_p^2 = .739$. Crucially, the interaction between Type of Stimulus and Type of Change was significant, $F(2, 140) = 3.87$, $MSE = 0.52$, $p = .023$, $\eta_p^2 = .053$, with the greatest transposition costs being seen for letter strings followed by symbol strings and finally for digit strings (Letter vs. Digit: $F(1, 70) = 5.45$, $MSE = 0.77$, $p = .022$, $\eta_p^2 = .072$; Letter vs. Symbol: $F(1, 70) = 5.71$, $MSE = 0.80$, $p = .019$, $\eta_p^2 = .076$; Digit vs. Symbol: $F(1, 70) < 0.1$, $p > .1$). There was also a significant interaction between Target Location and Type of Change, $F(1, 70) = 6.44$, $MSE = 0.62$, $p = .023$, $\eta_p^2 = .084$, with greater transposition costs when targets were displaced. The effect of Type of Change was nevertheless significant for both central targets ($F(1, 70) = 126.56$, $MSE = 19.15$, $p < .001$, $\eta_p^2 = .644$) and displaced targets ($F(1, 70) = 157.41$, $MSE = 30.16$, $p < .001$, $\eta_p^2 = .692$). No other interactions were significant.

## Discussion

Confirming prior work with the same-different matching task, transposition costs (i.e., smaller d' values in the transposition condition relative to the substitution condition, see Fig 4) were significantly larger for letter strings compared with both digit and symbol strings. Detecting a transposition change was harder than detecting a substitution change, and with a greater difficulty for letter strings as compared to other kinds of character strings. Target Location, on the other hand, had the same impact on same-different judgments to the three types of stimuli. Target displacement made the task harder for all types of stimuli, and caused an increase in transposition costs, again for all types of stimuli. We interpret these findings as revealing a common effect of target displacement on processing at the level of gaze-centered complex features (Fig 1). Randomly displacing target stimuli from trial-to-trial increases positional noise at this level of processing, hence increasing transposition costs for all three types of stimuli. As with our prior work, the greater overall transposition costs seen with letter stimuli are interpreted as reflecting the impact of a letter-specific, location-invariant order-encoding mechanism.

## Experiment 2

In Experiment 2, both reference and target stimuli were presented centrally, but the target stimulus was presented either with normal spacing or with an extra space between each of its constituent characters (e.g., DKLNFT–DNLKFT vs. DKLNFT–D N L K F T). Here one might

expect an increase in inter-character spacing to reduce positional noise in the processing of complex features, hence leading to reduced transposition costs (i.e., the opposite of the effects of the target displacement manipulation in Experiment 1). However, we acknowledge that this might trade-off with the variability of inter-character spacing from trial-to-trial increasing positional uncertainty when processing gaze-centered information. That is, although positional information might be more precise when inter-character spacing is increased (cf. the overlap model [16]), the fact that the spacing could vary from trial-to-trial could lead to an increase in positional uncertainty. Thus, as concerns the generic encoding of positional information, these two influences of the inter-character spacing manipulation on transposition effects might cancel each other out. Whatever the outcome of this trade-off, the key prediction is that the resulting effects should be the same for all three types of stimuli. Crucially, however, an increase in inter-character spacing is also hypothesized to increase the robustness of the open-bigram code for letter order information which therefore should increase the impact of open-bigram coding on transposed-letter effects relative to other transposed-character effects. In sum, although the overall impact of an increase in target inter-character spacing on transposition costs across all types of stimuli is not easy to predict, we do predict that letter-specific order encoding should benefit more from this manipulation than digits or symbols, and therefore that the greater transposition effects for letters should be further increased with larger inter-character spacing.

## Methods

**Participants.**   A total of 120 participants (71 female) with a mean age of 26 (SD = 9.4) years were recruited via the Prolific platform [45]. All of them were native speakers of French and had normal or corrected-to-normal vision. Prior the beginning of the experiment, participants were informed that data would be collected anonymously, and they provided informed consent before the experiment was initiated.

**Materials.**   The same materials as in Experiment 1 were used again in this experiment. Note, however, that the symbol "<" was replaced with the symbol "£" due to problems that arose with the LabVanced protocol used for running Experiment 2 [46]. Hence, 320 reference-target pairs for each Type of Stimulus (i.e., letter, digit, symbol) were used with the exact same manipulation of Type-of-Change (i.e., transposed vs. substituted characters). The only difference with respect to Experiment 1 was that instead of manipulating target displacement, here we manipulated the inter-character spacing of target stimuli. Therefore, while reference stimuli were always presented with normal spacing, half of the target stimuli were presented with normal spacing (i.e., 40 trials per category, e.g., DKLNFT–DNLKFT) and the other half had an extra space between each of the constituent characters (i.e., 40 trials per category, e.g., DKLNFT–D N L K F T). Note that to control for the size of the inter-character spacing, we used Courier New font which is a monospaced font (as used in Experiment 1). Therefore, all characters occupied exactly the same space on the screen (i.e., "K", "1" and "?" occupied the same space on screen) and the inter-character spacing corresponded exactly to the space occupied by one character. A 3x2x2 within-participants factorial design was constructed, with Type of Stimulus (letter, digit, symbol), Spacing (i.e., normal vs. spaced) and Type of Change (transposition vs. substitution) as main factors.

**Procedure.**   Participants performed this task on-line. The experiment was created using LabVanced [46] and displayed on the personal computer screen of participants. Before the experiment, we informed participants that the data would be collected anonymously, and we asked participants to select their gender and their age. Then the experimental instructions were presented on the screen. The procedure of Experiment 2 mimics the procedure used in

the previous experiment. That is, all stimuli were presented in Courier New font (size 16 pt.) in white on a black background. Thus, the different character strings occupied the same space on the screen whatever the type of character (letter, digit, symbol) and importantly, the distance between characters in a string was the same for all types of character in the normal spacing and extra inter-character spacing conditions. The procedure for each trial was the same as in Experiment 1 (see Fig 2), with first a fixation cross (500 ms), following by a reference stimulus (300 ms, horizontally centered and positioned 3mm above the exact center of the screen) and then the target stimulus (300 ms, horizontally centered and positioned 3mm below the center of the screen). Each trial ended with the participant's response, following by a 500 ms inter-trial interval. Participants responded "same" by pressing the "L" button on their computer keyboard and "different" by pressing the "S" button. After a short practice session, the main experiment started. Stimulus presentation was randomized and spilt into six blocks, that only included stimuli belonging to the same type of character.

## Results

As for Experiment 1, statistical analyses were performed only over the "different" trials (see S2 File for the complete set of data and additional analyses). The data from 23 participants were excluded prior to analysis (15 participants did not complete the experiment, 3 participants had an overall error rate higher than 70% on the critical "different" response trials, and 5 participants had a mean average RT greater than 1000 ms). The remaining sample was composed of 97 participants.

Similar to Experiment 1, error rates and participants' discriminability indices (d') were analyzed. As for the error data, the GLME model included Type of Character (letter, digit, symbol), Space (normal vs. spaced) and Type of Change (transposition vs. substitution) as fixed-factors. The models were fitted with the glmer function from the lme4 package [41] in the R statistical computing environment (version 4.1.0 [42]). ANOVAs were performed on d' indices with Type of Stimulus (letter, digit, symbol), Spacing (normal vs. spaced), and Type of Change (transposition vs. substitution) as main factors The Greenhouse and Geisser (1959) [43] correction was applied to determine the significance values for variables with more than two levels.

**Error rates.** The maximal random effects structure that converged was one including by-participant and by-item random intercepts. The following analyses were conducted taking the letter string condition as reference for the Type of Character factor, the normal condition as reference for the Spacing factor, and the substitution condition as reference for the Type of Change factor. We report regression coefficients (*b*), standard errors (*SE*) and z-values. Fixed effects were deemed significant if $|z| > 1.96$ [39]. Percentage of errors for each of the experimental conditions are presented in Fig 5.

Error rates were lower for digit strings than for letter strings (23.40% vs. 30.28% respectively; $b = 0.14$, $SE = 0.07$, $z = 0.03$), and lower for letter strings than for symbol strings (30.28% vs. 38.11% respectively; $b = -0.89$, $SE = 0.06$, $z < 0.01$). Fewer errors were observed when the target was presented with regular spacing compared with extra-spacing (29.77% vs. 31.42% respectively; $b = -0.15$, $SE = 0.05$, $z < 0.01$). There was also a significant effect of Type of Change, with fewer errors in the substitution condition than in the transposition condition (24.46% vs. 36.74% respectively; $b = -0.96$, $SE = 0.05$, $z < 0.01$). The interaction between Type of Change and Spacing was significant ($b = -0.17$, $SE = 0.07$, $z = 0.02$), reflecting the fact that transposition effects were greater in the extra-spacing condition (see Fig 5). Nevertheless, transposition effects were significant in both the normal spacing condition ($\chi^2(1) = 422.84$, $p < .001$) and the extra-spacing condition ($\chi^2(1) = 541.18$, $p < .001$). Transposition effects were

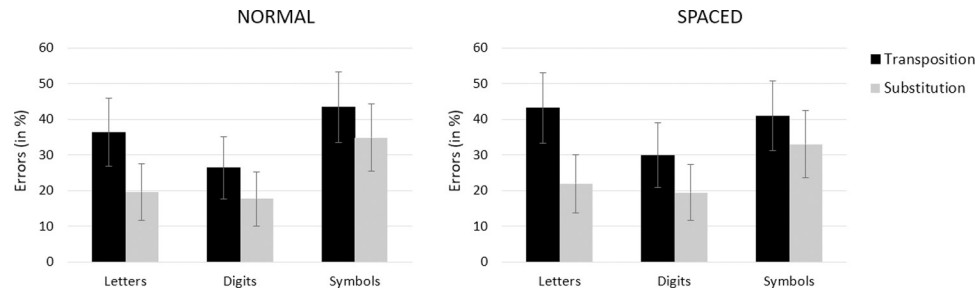

**Fig 5. Error rates for each type of stimulus in the transposition and in the substitution conditions when the target string was presented with normal spacing (left panel) and when it was presented with an extra space between characters in target stimuli (right panel) in Experiment 2.** Error bars represent within-participant 95% CIs [44].

significantly greater for letters compared with digits ($b = 0.38$, $SE = 0.08$, $z < 0.01$), and also greater for letters compared with symbols ($b = 0.54$, $SE = 0.07$, $z < .001$), although significant transposition effects were observed for each type of character (letters: $\chi^2(1) = 731.14$, $p < .001$, digits: $\chi^2(1) = 223.77$, $p < .001$, symbols: $\chi^2(1) = 129.72$, $p < .001$). The only other interaction to reach significance was the Type of Character x Spacing interaction when limited to the contrast between letters and symbols ($b = 0.25$, $SE = 0.07$, $z = 0.01$), with a greater interfering effect of extra-spacing for letters.

**Discriminability (d').** Condition means are presented in Fig 6. The analysis of d' values revealed significant main effects of Type of Stimulus, $F(2, 192) = 18.55$, $MSE = 71.76$, $p < .001$, $\eta_p^2 = .659$, Spacing, $F(1, 96) = 156.91$, $MSE = 49.29$, $p < .001$, $\eta_p^2 = .620$, and Type of Change, $F(1, 96) = 58.68$, $MSE = 14.27$, $p < .001$, $\eta_p^2 = .379$. Discriminability was poorest for symbols, then letters, and then digits—was lower overall with increased inter-character spacing—and standard transposition costs were observed (lower discriminability in the transposed-character condition compared with the substituted-character condition). The critical interaction between Type of Stimulus and Type of Change was again significant, $F(2, 192) = 4.26$, $MSE = 0.56$, $p = .015$, $\eta_p^2 = .043$, with greater transposition costs being seen for letters, although the effects were significant for all three types of character (letters: $F(1, 96) = 123.18$, $MSE = 37.65$, $p < .001$, $\eta_p^2 = .562$, digits: $F(1, 96) = 69.93$, $MSE = 13.95$, $p < .001$, $\eta_p^2 = .421$, symbols: $F(1, 96) = 40.03$, $MSE = 5.23$, $p < .001$, $\eta_p^2 = .294$). Most important is that the three-way interaction was significant, $F(2, 192) = 3.38$, $MSE = 0.37$, $p = .036$, $\eta_p^2 = .034$. Follow-up analyses revealed that the Spacing x Type of Change interaction was only significant for letter strings ($F(1, 96) = 6.27$, $MSE = 0.86$, $p = .014$, $\eta_p^2 = .061$), whereas it was marginal for digit strings ($F(1, 96) = 3.55$, $p = .063$) and did not reach significance for symbol strings ($F(1, 96) = $

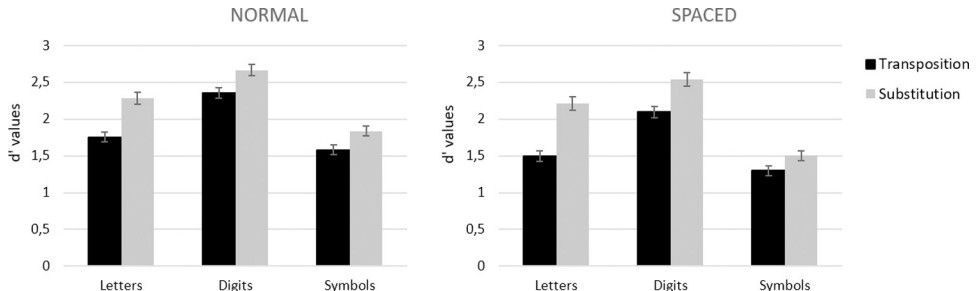

**Fig 6. d prime values for each type of stimulus in the transposition and in the substitution conditions when the target string was presented with normal spacing (left panel) and when it was presented with an extra space between characters in target stimuli (right panel) in Experiment 2.** Error bars represent standard errors.

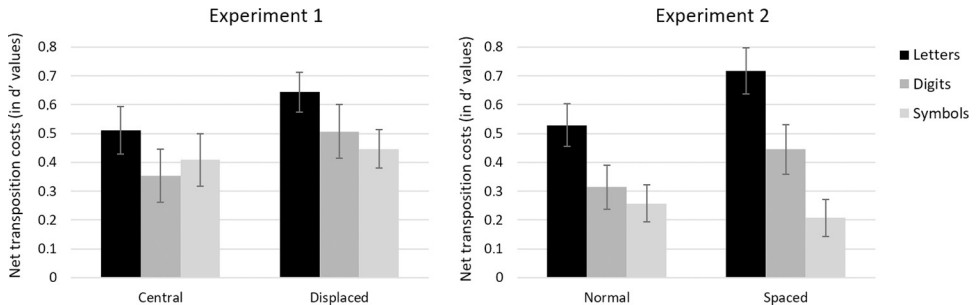

**Fig 7. Net transposition costs (in d' values) as a function of target displacement (Experiment 1, left panel) and inter-character spacing (Experiment 2, right panel) for the three types of stimuli.** Error bars represent standard errors.

0.51, $p > .1$). Transposition costs for letter stimuli were greater in the extra spacing condition than in the normal spacing condition (see Figs 6 and 7).

## Discussion

In Experiment 2, transposition costs were once again larger for letter strings relative to both digit and symbol strings. Adding an extra space between characters in the target stimulus made it harder to detect a change between reference and target (remember that reference stimuli were always presented with normal spacing). Furthermore, transposition costs for letter stimuli were significantly larger in the extra spacing condition than in the normal spacing condition in the d' measure. This paradoxical impact of increased inter-character spacing on transposed-letter effects can be explained by a more robust letter-order encoding in the extra spacing condition. We develop this explanation in the General Discussion.

## General discussion

The present study provided a further exploration of the sources of character-transposition effects in different types of character string (letters, digits, symbols) in the same-different matching task. In Experiment 1, the reference-target pairs could differ by transposing or substituting two non-contiguous characters. Moreover, whereas the reference string was always presented at the center of the screen, half of the targets were displayed in the center of the screen and the other half were shifted by two character positions to the left or to the right. Experiment 2 provided a further examination of the transposed-character effects by examining the influence of adding an extra space between characters in the target stimulus. In line with previous findings [3–5, 12, 13], the present results revealed that detecting a character transposition change was harder than detecting a character substitution change. Participants needed more time and made more errors for reference-target pairs containing character transpositions than for pairs containing character substitutions–a transposition cost. Furthermore, we replicated prior findings showing that the size of transposition costs was larger for letter strings than for both digit and symbol strings, which did not differ from each other [12, 13]. In line with some prior research [12, 13], we also found that same-different matching was easier for digit strings compared with both letter and symbol strings. This is possibly due to the fact that numbers form arbitrary combinations of digits, therefore conferring a greater familiarity to digit strings compared with both random consonant and symbol strings.

Over and above this replication of the now standard findings of greater transposition costs for letters compared with both digits and symbols in the same-different matching task, our experiments revealed two important new findings. First, in Experiment 1, same-different

judgments were harder to make when reference stimuli were randomly displaced horizontally relative to the reference stimulus, and this led to a statistically equivalent increase in transposition costs for all types of stimuli. In Experiment 2, on the other hand, although adding an extra space between characters in the target stimulus made the task harder, transposition costs were only found to significantly increase for letter stimuli. The overall pattern of transposition costs seen in both Experiments is summarized in Fig 7.

In the Introduction, we hypothesized that target displacement as well as extra spacing would add noise to the encoding of location-specific complex features, and this increase in positional noise would perturb the use of such location-specific features to establish a location-invariant ordering of each character identity in the string. Crucially, this increased positional noise affects the processing of letters, digits, and symbols alike, since they are all initially processed by gaze-centered complex feature detectors. This therefore accounts for the main effects of target location (Experiment 1) and of inter-character spacing (Experiment 2). Furthermore, at the second level of processing shown in Fig 1, a distinction is drawn between a generic mechanism for encoding location-invariant order information (left-hand panel) and a mechanism that is only used to represent letter order (right-hand panel). It is the existence of a letter-specific order encoding mechanism that accounts for the greater transposition costs for letter stimuli compared with both digits and symbols, as well as the fact that the extra spacing manipulation had a selective impact on transposition costs in d' values for letter stimuli, with a paradoxically greater transposition cost with increased inter-letter spacing. This paradoxical finding is explained by the more robust encoding of relative position information with letter stimuli in the extra spacing condition. More specifically, the increase in inter-letter spacing would enable a more accurate encoding of open-bigrams, which in our model are the source of letter-specific transposition effects. On the other hand, if transposed-character effects (including transposed-letter effects) were uniquely driven by positional noise [16], then one would expect an increase in inter-character spacing to systematically reduce the size of transposition effects.

It should nevertheless be mentioned that recent studies on reading have highlighted that the size of inter-letter spacing might have a different impact on word recognition depending on the font that is chosen and the precise amount of spacing that is implemented [47, 48]. Clearly, further work is needed to investigate the influence of the size of inter-character spacing in the same-different matching task. This could be done using a parametric experimental design in which the size of inter-character spacing would be manipulated (see [48]). Another line of future work, we have already launched, concerns the developmental trajectory of character transposition effects in same-different matching. Our model predicts that letter-specific effects should emerge and become stronger as reading expertise develops in children. Evidence that this might well be the case has been reported in studies investigating the development of transposed-letter effects using different paradigms (e.g., [34–36]).

In sum, we account for the present findings by assuming that the manipulations of target display (i.e., displacement or extra spacing between characters) increase generic positional noise at the level of complex feature detectors. Due to the generic nature of these mechanisms, this causes an increased difficulty in making same-different judgments for all types of character string. However, on top of these generic effects, a letter-specific mechanism for the location-invariant encoding of order information causes an overall increase in transposition costs for letter strings compared with the two other types of character strings and leads to a more robust encoding of letter order with greater inter-character spacing.

## Supporting information

**S1 File. Analyses performed on the RTs in Experiment 1.**
(PDF)

**S2 File. Analyses performed on the RTs in Experiment 2.**
(PDF)

## Author Contributions

**Conceptualization:** Stéphanie Massol, Jonathan Grainger.

**Data curation:** Stéphanie Massol.

**Formal analysis:** Stéphanie Massol.

**Methodology:** Stéphanie Massol, Jonathan Grainger.

**Writing – original draft:** Stéphanie Massol, Jonathan Grainger.

**Writing – review & editing:** Stéphanie Massol, Jonathan Grainger.

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
