## [Decision Letter · Decision Letter 0]

20 Dec 2021

PONE-D-21-35699Effects of horizontal displacement and inter-character spacing on transposed-character effects in same-different matchingPLOS ONE

Dear Dr. Massol,

Thank you for submitting your manuscript to PlosOne. I have now received two reviews from experts in your field.  As you will see from their comments below, both reviewers saw considerable merit in your paper, but both have also made some very constructive suggestions for revision.

At this stage, I am inviting you to submit a revised version of the paper for consideration.  In doing so, please carefully address each of the reviewers' comments and outline the changes you have made in a cover letter to me.

We look forward to receiving your revised manuscript.

Kind regards,

Francesca Peressotti, Ph.D

Academic Editor

PLOS ONE

Journal Requirements:

"JG was supported by ERC Grant 742141 "

Reviewers' comments:

Reviewer's Responses to Questions

**Comments to the Author**

1. Is the manuscript technically sound, and do the data support the conclusions?

Reviewer #1: Partly

Reviewer #2: Yes

2. Has the statistical analysis been performed appropriately and rigorously? 

Reviewer #1: Yes

Reviewer #2: Yes

3. Have the authors made all data underlying the findings in their manuscript fully available?

Reviewer #1: Yes

Reviewer #2: Yes

4. Is the manuscript presented in an intelligible fashion and written in standard English?

Reviewer #1: Yes

Reviewer #2: Yes

5. Review Comments to the Author

Reviewer #1: Review of ms “Effects of horizontal displacement and inter-character spacing on transposed-character effects in same-different matching”

This manuscript presents two experiments with a same-different matching task comparing letter strings, digit strings, and symbol strings—at the same time there is also a manipulation of displacement and inter-character spacing.

I believe that this is a nice manuscript and it should be published after some revision. This is a list of issues that need some work in a revised version of the manuscript:

(1) While not the most important issue, I am not sure the Procedure of Experiment 2 (the online experiment) was identical to that of Experiment 1 (as stated in the ms). Surely, Presentation was not used in Experiment 2, and perhaps some other details also differed. The exact info should be provided.

(2) The authors should justify in the Introduction why an SDT analysis was going to be chosen and why—in particular because the RT is typically the gold-standard for dependent variables in most tasks (and in prior experiments). The authors did report the standard RT and accuracy analyses in the Supplemental Materials, but there was no interaction of type of stimulus in the RT analyses of Experiment 1. The accuracy data showed a relatively weak interaction (Letters vs. Symbols in Experiment 1). Also, some writing of the findings would help (i.e., not just reporting the Tables) and this would include an account of the similarities and differences of the transposition effect for the three types of stimuli across the two experiments.

(3) I’m not sure ANOVAs are the best method to deal with SDTs. I believe that there are currently packages to run linear mixed effects models on SDT measures. Given that the critical interaction (transposition effect for the three different types of stimuli) was not overwhelming, I think readers would like to know not just whether the interaction is significant, but also whether we can use the data to critically favor one model of serial order over others.

(4) Related to the interpretive issues above, I believe that there is some agreement that neither pure “open bigrams” nor pure “positional noise” models alone can offer an account of the data. To cite just one example: Highlighting the critical letters and other perceptual manipulations reduce the transposed-letter effect (e.g, doi: 10.1177/1747021818789876I) and this pattern cannot be easily accommodated by any strong version of the models. Clearly, at some point in processing, the way letters are processed must differ from symbols and digits.

(5) The present experiment used an unprimed paradigm that cannot tell apart whether the differences occurred early or late during processing. Using a masked priming procedure applied to the same-different task, García-Orza et al. (2010, doi: 10.1080/17470210903474278) showed that masked transposed-character effects were similar in magnitude for letter strings, symbol strings, and digit strings. If the different paths for these types of stimuli had occurred very early in processing, one would have expected quite different effects for letter strings, symbol strings, and digit strings. This needs to be acknowledged in the manuscript—note that this pattern is more consistent with “position uncertainty” models.

(5) Some more information on the characteristics of inter-character spacing would be welcome. Inter-character spacing differs across fonts. Indeed, there are some fonts that mainly differ in their inter-character spacing (e.g., Tahoma vs. Calibri) and this spacing has an influence on word recognition and reading (see the following JEPLMC paper: doi: 10.1037/xlm0000477 and the following JEP:Applied paper 10.1037/xap0000104). Also, in Experiment 2, as it was online, there was no clear control on the distance from the eyes to the screen. Thus, while the findings of inter-character spacing are suggestive (e.g., for further eyetracking work), I believe the authors should be cautious—further, the manipulation was not parametric and one might argue that the pattern of data would be different with a subtler manipulation (i.e., a bit of extra inter-letter spacing may help, but not beyond a certain point, and there are a number of papers published on this issue).

(6) Finally, the authors may want to discuss why accuracy (or SDT) is more sensitive to the transposed-character effects across letters, digits and symbols than the response times. Typically, we all assume that these are two sides of the same coin, but probably things are more complicated.

In sum, this is a nice manuscript that needs some polishing to strengthen the main arguments.

Reviewer #2: SUMMARY ARTICLE PONE-D-21-35699. The authors report two same-different matching experiments in which subjects are exposed to either letter, digit or symbol strings in distinct visual constraint conditions. In Experiment 1, targets are presented either in the center of shifted to the right or left with respect to the reference. In experiment 2, authors tested whether learning was modulated by procedural or declarative memory abilities. Targets are presented centered but with increased inter-character spacing. Signal detection -d´- results point to a greater detection cost in the transposed condition compared to the substitution condition, and although target displacement and inter-character spacing was detrimental for all conditions, a greater detection cost was observed for transposed letter strings than for symbol or digit strings. Findings are interpreted in the light of a two level position coding mechanism (location specific level and location invariant level either generic or specific for letters)

I like the topic and the way the article is written. Position coding experiments can shed light on the mechanisms involved in letter and orthographic processing, and on the constraints underlying such processes. I think the article has the potential to provide such information but there are some issues that in my opinion should be tackled before publication.

First, description of the model. The authors describe a letter position coding model that accounts for the data observed and that seems suitable to predict and explain the obtained results. However it would be suitable that they also describe other models and explain the differences between their model and other letter position models (although they mention the overlap model, this is not deeply describe, nor the processing mechanisms that account for the different predictions between models). Additionally it would be suitable that the authors explain why this model accounts for an specific position invariant coding mechanism for letters, and how this fact can be linked to the “long term memory orthographic knowledge” claim. Does the model account for this knowledge based coding? Can this specific coding level be accounted in terms of visual experience? This should be better explained in the introduction.

Second, better justification of the task selection. The same-different matching task seems appropriate to test letter coding processes but it taps very early perceptual processes. Why was this task was selected? Why not a lexical decision one? Why is this task specifically interesting to measure coding processes? Before experiment 1 the task is described but it deserves a better justification, particularly to contextualize the data and the implications within the process of word recognition and/or reading.

Third, consideration upon the limitations of the study and proposals for future studies. Before the last paragraph of the discussion limitations and proposals for future research should be included. The fact that a same-different task was employed provides data about very early perceptual processes involved in word recognition. What can this inform orthographic learning processes? What should be the next step? What could other tasks/ages/studies add to the field of orthographic processing and reading?

I think that these issues should be better considered for the article to have a stronger impact. However, I think that if the suggested questions are examined in more depth it could be perfectly published in PONE.

6. PLOS authors have the option to publish the peer review history of their article (what does this mean?). If published, this will include your full peer review and any attached files.

Reviewer #1: No

Reviewer #2: No

---

## [Author Response · Author response to Decision Letter 0]

16 Feb 2022

Dear Editor,

We would like to thank you and the reviewers for the evaluation and the constructive criticism of our manuscript entitled " Effects of horizontal displacement and inter-character spacing on transposed-character effects in same-different matching" (Submission ID PONE-D-21-35699) submitted to Plos One. We are very happy to be able to submit a revision that takes into consideration all the points raised by the reviewers. The detailed responses are provided below in bold after each point that was raised. All changes are indicated in red color in the revised manuscript.

Yours sincerely,

Stéphanie Massol and Jonathan Grainger

Reviewer #1: Review of ms “Effects of horizontal displacement and inter-character spacing on transposed-character effects in same-different matching”

This manuscript presents two experiments with a same-different matching task comparing letter strings, digit strings, and symbol strings—at the same time there is also a manipulation of displacement and inter-character spacing.

I believe that this is a nice manuscript and it should be published after some revision. This is a list of issues that need some work in a revised version of the manuscript:

Response: We thank the reviewer for the overall positive evaluation of our work.

(1) While not the most important issue, I am not sure the Procedure of Experiment 2 (the online experiment) was identical to that of Experiment 1 (as stated in the ms). Surely, Presentation was not used in Experiment 2, and perhaps some other details also differed. The exact info should be provided.

Response: We thank the reviewer for raising this point. We now state that this experiment was run on-line using LabVanced. As we already said, the procedure of this experiment mimics the procedure used in Experiment 1. Nevertheless, more detailed information about the procedure is now provided.

(2) The authors should justify in the Introduction why an SDT analysis was going to be chosen and why—in particular because the RT is typically the gold-standard for dependent variables in most tasks (and in prior experiments). The authors did report the standard RT and accuracy analyses in the Supplemental Materials, but there was no interaction of type of stimulus in the RT analyses of Experiment 1. The accuracy data showed a relatively weak interaction (Letters vs. Symbols in Experiment 1). Also, some writing of the findings would help (i.e., not just reporting the Tables) and this would include an account of the similarities and differences of the transposition effect for the three types of stimuli across the two experiments.

Response: We now justify our choice of using SDT for data analyses at the end of the Introduction. It should be noted that the procedure we used, with a brief presentation duration of the reference stimulus, leads to quite high error rates (see also Duñabeitia et al., 2012; Massol et al., 2013). Therefore, error rates could be considered to be the main dependent variable. Indeed, when errors are encouraged (by short stimuli durations), as in this investigation, then it could be argued that RTs are no longer that reliable as a dependent variable. We nevertheless analyze RTs because the task required participants to respond as rapidly as possible (see response to point 6 below for more information). We now present the findings based on analyses performed on the error rates in the manuscript (see also response to the following comment) and the results based on RTs are now fully reported in the Supplemental Materials. We have also tried to improve the rerporting of our findings and we provide a more detailed account of the similarities and differences of transposition effects for the three types of stimuli in the General Discussion.

(3) I’m not sure ANOVAs are the best method to deal with SDTs. I believe that there are currently packages to run linear mixed effects models on SDT measures. Given that the critical interaction (transposition effect for the three different types of stimuli) was not overwhelming, I think readers would like to know not just whether the interaction is significant, but also whether we can use the data to critically favor one model of serial order over others.

Response: We report ANOVAs on discriminability indices because to our knowledge there is no established (at least not published) method for dealing with SDT measures using mixed-effects models. However, in order to provide complementary statistical analyses, we now present the results from the glme analyses performed on error rates in the main manuscript (RT analyses are presented in the Supplemental Materials).

(4) Related to the interpretive issues above, I believe that there is some agreement that neither pure “open bigrams” nor pure “positional noise” models alone can offer an account of the data. To cite just one example: Highlighting the critical letters and other perceptual manipulations reduce the transposed-letter effect (e.g, doi: 10.1177/1747021818789876I) and this pattern cannot be easily accommodated by any strong version of the models. Clearly, at some point in processing, the way letters are processed must differ from symbols and digits.

Response: We totally agree with the reviewer here. We now discuss the study that is referenced in the comment and explain why any strong version of these two approaches (perceptual-based vs. orthographic representation-based) cannot capture the whole pattern of results. However, we also stress the fact that our model incorporates both mechanisms, such that letters, digits, and symbols are indeed processed in the same way in early processing, and that differences between stimulus types emerge later in processing. This is discussed in more detail in the Introduction and General Discussion sections in the revision.

(5) The present experiment used an unprimed paradigm that cannot tell apart whether the differences occurred early or late during processing. Using a masked priming procedure applied to the same-different task, García-Orza et al. (2010, doi: 10.1080/17470210903474278) showed that masked transposed-character effects were similar in magnitude for letter strings, symbol strings, and digit strings. If the different paths for these types of stimuli had occurred very early in processing, one would have expected quite different effects for letter strings, symbol strings, and digit strings. This needs to be acknowledged in the manuscript—note that this pattern is more consistent with “position uncertainty” models.

Response: We now mention this study when describing the Overlap model (Gomez et al., 2008) in the Introduction: “In support of this approach, when using the masked priming version of the same-different matching task, García-Orza, Perea and Muñoz (2010) found a highly similar transposed-character effect for letter, digit and symbol strings (see also Garcia-Orza, Perea, & Estudillo, 2011).” We would nevertheless point out that adding masked priming to the same-different task adds an extra level of complication when interpreting effects.

(5) Some more information on the characteristics of inter-character spacing would be welcome. Inter-character spacing differs across fonts. Indeed, there are some fonts that mainly differ in their inter-character spacing (e.g., Tahoma vs. Calibri) and this spacing has an influence on word recognition and reading (see the following JEPLMC paper: doi: 10.1037/xlm0000477 and the following JEP:Applied paper 10.1037/xap0000104). Also, in Experiment 2, as it was online, there was no clear control on the distance from the eyes to the screen. Thus, while the findings of inter-character spacing are suggestive (e.g., for further eyetracking work), I believe the authors should be cautious—further, the manipulation was not parametric and one might argue that the pattern of data would be different with a subtler manipulation (i.e., a bit of extra inter-letter spacing may help, but not beyond a certain point, and there are a number of papers published on this issue).

Response: We have now added information regarding the characteristics of inter-character spacing in the manuscript. Of course, we knew that inter-character spacing differs across fonts so we used Courrier New font which is a monospaced font. Therefore, in Experiment 2, the inter-character spacing corresponds exactly to the space that could be occupied by a character. However, we agree with the reviewer that, in Experiment 2, it was not possible to control the distance from the eyes to the screen, since the experiment was performed on-line and displayed on the personal computer screen of participants. We therefore acknowledge, in the General Discussion, that future work is needed with more precise and more subtle manipulations of inter-letter spacing.

(6) Finally, the authors may want to discuss why accuracy (or SDT) is more sensitive to the transposed-character effects across letters, digits and symbols than the response times. Typically, we all assume that these are two sides of the same coin, but probably things are more complicated.

Response: As mentioned above, in the same-different matching paradigm using a brief presentation duration of the reference stimuli, high error rates are typically observed (see also Duñabeitia et al., 2012; Massol et al., 2013), and therefore one could argue that error rates (or SDT analyses) should be considered to be the main dependent variable. However, since participants were requested to respond as rapidly and as accurately as possible, we also analyzed RTs (see Supplemental Materials). But we nevertheless insist on the fact that a significant effect in error rates is just as valid as an effect seen in RTs.

In sum, this is a nice manuscript that needs some polishing to strengthen the main arguments.

Response: We thank the reviewer for this positive evaluation of our work.

Reviewer #2: SUMMARY ARTICLE PONE-D-21-35699. The authors report two same-different matching experiments in which subjects are exposed to either letter, digit or symbol strings in distinct visual constraint conditions. In Experiment 1, targets are presented either in the center of shifted to the right or left with respect to the reference. In experiment 2, authors tested whether learning was modulated by procedural or declarative memory abilities. Targets are presented centered but with increased inter-character spacing. Signal detection -d´- results point to a greater detection cost in the transposed condition compared to the substitution condition, and although target displacement and inter-character spacing was detrimental for all conditions, a greater detection cost was observed for transposed letter strings than for symbol or digit strings. Findings are interpreted in the light of a two level position coding mechanism (location specific level and location invariant level either generic or specific for letters)

I like the topic and the way the article is written. Position coding experiments can shed light on the mechanisms involved in letter and orthographic processing, and on the constraints underlying such processes. I think the article has the potential to provide such information but there are some issues that in my opinion should be tackled before publication.

Response: We thank the reviewer for this overall quite positive evaluation.

First, description of the model. The authors describe a letter position coding model that accounts for the data observed and that seems suitable to predict and explain the obtained results. However it would be suitable that they also describe other models and explain the differences between their model and other letter position models (although they mention the overlap model, this is not deeply describe, nor the processing mechanisms that account for the different predictions between models). Additionally it would be suitable that the authors explain why this model accounts for an specific position invariant coding mechanism for letters, and how this fact can be linked to the “long term memory orthographic knowledge” claim. Does the model account for this knowledge based coding? Can this specific coding level be accounted in terms of visual experience? This should be better explained in the introduction.

Response: In the Introduction, we now provide a more detailed description of both our model and the overlap model which we consider to be good examples the two main classes of model. In particular, as requested, we provide more details concerning the link between location-specific perceptual processing and location-invariant order encoding in our model. Note that we have now dropped the reference to short-term and long-term memory which was a bit of a sidetrack, and we provide more information about the hypothesized development of the reading-specific mechanism, and why such a mechanism could be useful for skilled reading.

Second, better justification of the task selection. The same-different matching task seems appropriate to test letter coding processes but it taps very early perceptual processes. Why was this task was selected? Why not a lexical decision one? Why is this task specifically interesting to measure coding processes? Before experiment 1 the task is described but it deserves a better justification, particularly to contextualize the data and the implications within the process of word recognition and/or reading.

Response: We needed a task that could be applied to both letters and non-letter stimuli (symbols in this study), so lexical decision was obviously excluded. We now more clearly motivate the choice of the SD matching task in the Introduction. However, we do agree that it would be interesting to explore other tasks that could be applied to letter and non-letter stimuli (see response to following point).

Third, consideration upon the limitations of the study and proposals for future studies. Before the last paragraph of the discussion limitations and proposals for future research should be included. The fact that a same-different task was employed provides data about very early perceptual processes involved in word recognition. What can this inform orthographic learning processes? What should be the next step? What could other tasks/ages/studies add to the field of orthographic processing and reading?

Response: We now underline the fact, as exemplified in our model, that this study focused on early perceptual processes in reading and character processing in general, and we now tentatively point to future work that might tap higher-level processing by using different tasks. We note that concerning the issue of development of orthographic processing, we are in the process of running a study investigating the developmental trajectory of transposed-character effects in same-different matching in a large sample of primary school children (this project recently received funding from the French National Research Agency (ANR) – awarded to Stéphanie Massol). This work in progress in now mentioned at the end of the General Discussion when mentioning future lines of research.

I think that these issues should be better considered for the article to have a stronger impact. However, I think that if the suggested questions are examined in more depth it could be perfectly published in PONE.

Response: We again thank this reviewer for the positive evaluation of our work.

---

## [Decision Letter · Decision Letter 1]

2 Mar 2022

Effects of horizontal displacement and inter-character spacing on transposed-character effects in same-different matching

PONE-D-21-35699R1

Dear Dr. Massol,

We’re pleased to inform you that your manuscript has been judged scientifically suitable for publication and will be formally accepted for publication once it meets all outstanding technical requirements.

Kind regards,

Francesca Peressotti, Ph.D

Academic Editor

PLOS ONE

Additional Editor Comments (optional):

Dear Authors,

It is a pleasure to accept your manuscript entitled "Effects of horizontal displacement and inter-character spacing on transposed-character effects in same-different matching?" in its current form for publication in PlosOne. The comments of the reviewers who reviewed your manuscript are included at the foot of this letter.

Reviewer 1 helpfully pointed out a couple of possible typos / remaining small points of clarification that you may wish to address as you prepare the manuscript to be publication ready.

Thank you for your fine contribution.

Best whishes

Francesca

Reviewers' comments:

Reviewer's Responses to Questions

**Comments to the Author**

1. If the authors have adequately addressed your comments raised in a previous round of review and you feel that this manuscript is now acceptable for publication, you may indicate that here to bypass the “Comments to the Author” section, enter your conflict of interest statement in the “Confidential to Editor” section, and submit your "Accept" recommendation.

Reviewer #1: All comments have been addressed

Reviewer #2: All comments have been addressed

2. Is the manuscript technically sound, and do the data support the conclusions?

Reviewer #1: Yes

Reviewer #2: Yes

3. Has the statistical analysis been performed appropriately and rigorously? 

Reviewer #1: Yes

Reviewer #2: Yes

4. Have the authors made all data underlying the findings in their manuscript fully available?

Reviewer #1: Yes

Reviewer #2: Yes

5. Is the manuscript presented in an intelligible fashion and written in standard English?

Reviewer #1: Yes

Reviewer #2: Yes

6. Review Comments to the Author

Reviewer #1: I think the authors did a very good job. I only have a couple of formal comments for the “proofing” stage: (1) the reference of the R program was from 2018, but surely these analyses were conducted in 2021 or so, so I would update the year; (2) some of the references do not follow the rules of Plos One—this should be very easy to fix.

Reviewer #2: Te authors have adequately responded to all the queries made in the review. My greatest concerns were related to the justification of the model's claims regarding coarse grained coding due to experience with letters, and of the task used to measure the processes under interest. This two issues are better focused now, and the paper is much better articulated.

7. PLOS authors have the option to publish the peer review history of their article (what does this mean?). If published, this will include your full peer review and any attached files.

Reviewer #1: No

Reviewer #2: No

---

## [Editor Report · Acceptance letter]

11 Mar 2022

PONE-D-21-35699R1 

Effects of horizontal displacement and inter-character spacing on transposed-character effects in same-different matching 

Dear Dr. Massol:

I'm pleased to inform you that your manuscript has been deemed suitable for publication in PLOS ONE. Congratulations! Your manuscript is now with our production department. 

Kind regards, 

on behalf of

Dr. Francesca Peressotti 

Academic Editor

PLOS ONE